# Personalized Neural Architecture Search for Federated Learning

## Abstract

Federated Learning (FL) is a recently proposed learning paradigm for decentralized devices to collaboratively train a predictive model without exchanging private data. Existing FL frameworks, however, assume a one-size-fit-all model architecture to be collectively trained by local devices, which is determined prior to observing their data. Even with good engineering acumen, this often falls apart when local tasks are different and require diverging choices of architecture modelling to learn effectively. This motivates us to develop a novel personalized neural architecture search (NAS) algorithm for FL. Our algorithm, FEDPNAS, learns a base architecture that can be structurally personalized for quick adaptation to each local task. We empirically show that FEDPNAS significantly outperforms other NAS and FL benchmarks on several real-world datasets.

## 1 Introduction

Federated Learning (FL) (McMahan et al., 2017) is a variant of distributed learning where the objective function can be decomposed into a linear combination of $M$ local objective functions. Each function depends on its private data hosted by a local client and a set of shared parameters $w$,

$$\operatorname*{argmin}_{w} \mathcal{L}(w) \quad \equiv \quad \operatorname*{argmin}_{w} \sum_{i=1}^{M} \mathcal{L}_i(w \mid \mathcal{D}_i) , \tag{1}$$

where $\mathcal{D}_i$ denotes the $i^{\text{th}}$ local training dataset comprising input-output tuples $(x, y)$. In a standard supervised learning task where the predictive model is modeled as a fixed deep neural network $\psi$ with learnable weights $w$, let $\ell(x, y)$ denote the loss incurred by predicting $\psi(x; w)$ when the true output is $y$. The expected loss of $\psi(x; w)$ on $\mathcal{D}_i$ is given as

$$\mathcal{L}_i(w \mid \mathcal{D}_i) \quad = \quad \mathbb{E}_{(x,y) \sim \mathcal{D}_i} \Big[ \ell(x, y; \psi) \Big] . \tag{2}$$

This is not applicable to scenarios where local models are expected to solve different tasks which are similar in broad sense yet diverge in finer details. For example, consider the task of recognizing the outcome of a coin flip given images collected by two clients: one capture the coin from above, the other from below. This setting implies that when the same input image is provided by both clients, the correct classifications must be the opposite of one another. However, since existing FL methods converge on a single model architecture and weight, there can only be one predictive outcome which cannot satisfy both tasks.

To relax this constraint, the recent work of Fallah et al. (2020) extends FL by incorporating ideas from meta learning (Finn et al., 2017) which results in a new framework of personalized FL. The new framework can accommodate for such task heterogeneity but still requires all client models to agree on a single architecture beforehand, which is sub-optimal. To address this shortcoming, one naive idea is to adopt existing ideas in Neural Architecture Search (NAS) via Reinforcement Learning (Zoph and Le, 2016; Pham et al., 2018) which act as an outer loop to the existing FL routine.

However, this simple approach does not allow client models to adapt to local tasks on an architecture level and is often not preferred due to the cost of repeated FL training. This paper proposes a novel personalized NAS algorithm for federated learning, which generalizes ideas in respective areas of NAS (Zoph and Le, 2016; Pham et al., 2018) originally developed for single-task scenarios, and

FL (Fallah et al., 2020) under a unified len of federated personalized neural architecture search (FEDPNAS).

In particular, to customize the model architecture for each task in the FL workflow, FEDPNAS first represents the model architecture for each task as a sub-network sampled from a large, over-parameterized network. The sampling distribution is (collaboratively) learned along with the parameters of the sampled network via a generalization of the recently proposed Discrete Stochastic NAS (DSNAS) method (Hu et al., 2020). Unlike DSNAS, which lacks the ability to customize architecture for individual tasks, our generalized FEDPNAS incorporates model personalization on an architecture level. Our contributions include:

**1.** A novel architecture that factorizes into a base component (shared across tasks) and a personalizable component, which respectively capture the task-agnostic and task-specific information (Section 3.2).

**2.** A context-aware sampling distribution conditioned on specific task instance, which captures task-specific information and naturally incorporates personalization into architecture search (Section 3.4).

**3.** An FL algorithm that optimizes for a common architecture, followed by a personalization phase where each client subsequently adapts only the *personalized component* to fit its own task via fine-tuning with local data (Section 3.1). To ensure that the common architecture distribution converges at a *vantage* point that is relevant and beneficial to all clients, we generalize the vanilla FL objective in Eq. equation 1 such that local gradient steps directly optimize for expected improvement resulting from future fine-tuning (Section 3.3).

**4.** A theoretical perspective on our FL objective (Section 3.5 and thorough empirical analysis showing significant performance gain compared to state-of-the-art FL and NAS methods (Section 4).

## 2  RELATED WORKS

### 2.1  TWO-STAGE NEURAL ARCHITECTURE SEARCH

Most existing NAS frameworks separately optimize for the optimal architecture and its parameters in two stages: *searching* and *evaluation*. The former stage usually employs evolutionary-based strategies (Floreano et al., 2008; Real et al., 2019), Bayesian optimization surrogates (Bergstra et al., 2013; Hu et al., 2018) or Reinforcement Learning controllers (Baker et al., 2016; Zoph and Le, 2016; Pham et al., 2018) to propose candidate architectures based on random mutations and/or observed experience; while the latter optimizes the parameters of these architectures given task data and provide feedback to improve the search agent. Naturally, an extension of such methods to the FL setting is through distributing the evaluation workload over many clients, which does not require exposing private data. In practice, however, two-stage federated NAS frameworks are generally not suitable for the personalized FL setting for two reasons: (a) the clients often lack the computational capacity to repeatedly optimize the parameters for many candidate architectures; and (b) the clients have to converge on a single architecture proposed by the central search agent.

### 2.2  DISCRETE STOCHASTIC NEURAL ARCHITECTURE SEARCH

Discrete stochastic neural architecture search (DSNAS) (Hu et al., 2020) addresses the computational issue of two-stage NAS by jointly optimizing the optimal architecture and its weight in an end-to-end fashion, which allows users to continually train a single network on demand over time as opposed to performing full parameter optimization for every candidate until a good architecture is discovered.

The main idea of DSNAS is to combine weight training for an over-parameterized master architecture with discrete computational path sampling. DSNAS parameterizes the master architecture as a stack of modular cells: $\boldsymbol{\psi}(x) = \psi_C \circ \psi_{C-1} \cdots \circ \psi_1\left(x\right)$[1], where $x$ is an arbitrary input, $C$ is the number of cells, $\psi_t$ denotes the $t^{\text{th}}$ cell in the stack, and $\circ$ denotes the compositional operator. The inner computation of $\psi_t$ is in turn characterized by a directed acyclic graph (DAG) with $V$ nodes $\{v_i\}_{i=1}^{|V|}$, where each node represents some intermediate feature map. For each directed edge $(v_i, v_j)$,

---

[1]Although each DSNAS cell receives the outputs of two previous cells as inputs, we simplify the number to one for ease of notation.

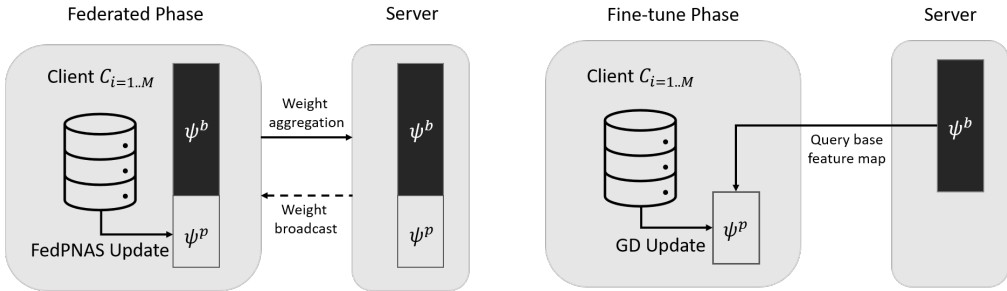

Figure 1: Our proposed method FEDPNAS consists of (1) a federated learning phase, where each client updates both the base component ($\psi^b$) and the personalized component ($\psi_p$) the architecture using the FEDPNAS update (Section 3.3) and sends its parameters to the central server for aggregation; and (2) a fine-tune phase, where each client updates only the personalized component of the architecture using standard gradient update.

there is an associated list of $D$ possible network operations $\mathbf{O}_{ij} = \left[o_{ij}^1, o_{ij}^2 \ldots o_{ij}^D\right]^2$ where each operation $o_{ij}^k$ transforms $v_i$ to $v_j$. Here, $v_1$ corresponds to the output of previous cell $\psi_{t-1}$ (or input $x$ when $t = 1$). We recursively define intermediate nodes $v_j = \sum_{i=1}^{j-1} \mathbf{Z}_{ij}^\top \mathbf{O}_{ij}(v_i)$, where $\mathbf{O}_{ij}(v_i) \triangleq \left[o_{ij}^1(v_i), o_{ij}^2(v_i) \ldots o_{ij}^D(v_i)\right]$ and $\mathbf{Z}_{ij}$ is a one-hot vector sampled from the categorical distribution $p(\mathbf{Z} \mid \mathbf{\Pi})$ where the event probabilities $\mathbf{\Pi} = \{\pi_1, \pi_2, \ldots, \pi_D \mid \sum_{i=1}^{D} \pi_i = 1\}$ are learnable. Essentially, learning the distribution $p(\mathbf{Z})$ allows us to sample computational paths or sub-graphs of the original DAG that correspond to high-performing, compact architecture from the over-parameterized master network. Sampling discrete random variables from $p(\mathbf{Z})$, however, does not result in a gradient amenable to back-propagation. To sidestep this issue, DSNAS adopts the *straight-through* Gumbel-softmax trick (Jang et al., 2016), which re-parameterizes the $k^{\text{th}}$ index of the one-hot variable as $\mathbf{Z}_{ij}[k] = \mathbb{I}\left(k \triangleq \arg\max_t \left[g_t + \log \pi_t\right]\right)$, where $g_t \sim \text{Gumbel}(0, 1)$. While this *forward* computation does not have a gradient by itself, we can estimate the gradient through a proxy during the *backward* pass:

$$\nabla \mathbf{Z}_{ij}[k] \simeq \nabla \tilde{\mathbf{Z}}_{ij}[k] \triangleq \nabla \left(\frac{\exp\left((g_k + \log \pi_k)/\tau\right)}{\sum_{t=1}^{D} \exp\left((g_t + \log \pi_t)/\tau\right)}\right) \tag{3}$$

which is unbiased when converged as the temperature $\tau$ is steadily annealed to 0 (Jang et al., 2016). This formulation, however, is not easily extended to the FL setting, especially when local tasks are not homogeneous. The key challenges in doing so are described in Section 3, together with our proposed approaches.

## 3 PERSONALIZED NAS FOR FEDERATED LEARNING

### 3.1 FEDERATED LEARNING OF DSNAS

Let $\mathbf{W}$ denote the concatenated weights of all network operations in the network architecture. The set up above of DSNAS (Jang et al., 2016) is then naïvely extendable to a FL setting via the following objective formulation:

$$\operatorname*{argmin}_{\mathbf{W},\mathbf{\Pi}} \mathcal{L}(\mathbf{W}, \mathbf{\Pi}) \equiv \operatorname*{argmin}_{\mathbf{W},\mathbf{\Pi}} \frac{1}{M} \sum_{i=1}^{M} \mathcal{L}_i(\mathbf{W}, \mathbf{\Pi} \mid \mathcal{D}_i). \tag{4}$$

McMahan et al. (2017) optimizes this objective by alternating between (a) central agent broadcasting aggregated weights to local clients and (b) local clients sending gradient descent updated weights (given local data) to the central agent for aggregation. This, however, implies that after the last

---

[2] We drop the cell index in the definition operation for ease of notation.

central aggregation step, all clients will follow the same architecture distribution induced by the final broadcasted copy of $\mathbf{W}$ and $\mathbf{\Pi}$. As previously argued, this is not optimal in a heterogenous task setting which requires task-specific adaptation for local clients to achieve good performance.

Furthermore, having the same sampling distribution $p(\mathbf{Z})$ regardless of context (i.e., feature maps received as cell input) limits the architecture discovery to those that perform reasonably on average over the entire dataset. However, we remark that restricting the architecture to be the same for every input samples is unnecessary and undermines the expressiveness of an over-parameterized search space. On the other hand, letting the architecture be determined on a *per-sample* basis makes better use of the search space and potentially improves the predictive performance.

The focus of this work is therefore to incorporate both task-wise and context-wise personalization to federated neural architecture search in multitask scenarios, which is achieved through our proposed algorithm FEDPNAS. In general, FEDPNAS functions similarly to the vanilla FEDDSNAS algorithm described above, with an addition of a fine-tuning phase at each local client after the FL phase to adapt the aggregated common model for local task data, as shown in Fig. 1. To make this work, however, we need to address the following **key challenges**:

**C1.** First, as previously argued in Section 1, tasks across federated clients tend to share similarities in broad sense, and diverge in finer details. A good federated personalization search space, therefore, need to capture this fundamental observation through design and appropriate resource distribution. We address this challenge in Section 3.2.

**C2.** Second, a major advantage of having an over-parameterized architecture search space is the flexibility of having specific computation paths for different samples, which is not exploited by DSNAS as reflected in its choice of context-independent sampling distribution $p(\mathbf{Z})$. To address this, Section 3.4 proposes a novel parameterization of $p(\mathbf{Z})$ to incorporate context information into operator sampling.

**C3.** Last, while the *fine-tuning* phase is designed to incorporate *task-personalization*, there is no guarantee that the common model can be quickly adapted to client tasks (Fallah et al., 2020). The common model may end up in a localization that favors one client over another, which makes it difficult for the latter to fine-tune. To address this concern, Section 3.3 proposes a new personalized federated NAS objective inspired by Finn et al. (2017) to optimize the common model in anticipation of further fine-tuning by the client models.

## 3.2 PERSONALIZABLE ARCHITECTURE SEARCH SPACE

Similar to DSNAS (Hu et al., 2020), our framework adopts a cell-based representation (Section 2.2) to trade-off search space expressiveness for efficiency, which is extremely suitable for FL where clients tend to have low-end computational capacity. Unlike the original design which assumes similar role for every cell in the architecture stack (i.e., as reflected by their choice of fully factorizable path sampling distribution $p(\mathbf{Z})$), we instead split our cell stack into two components with separate meta-roles catering to the federated personalization task: (a) a *base stack* $\boldsymbol{\psi}_{\mathrm{b}} = \{\psi_1^b, \psi_2^b \dots \psi_{C_b}^b\}$ which aims to capture the broad commonalities of data samples across client tasks; and (b) *personalized stack* $\boldsymbol{\psi}_{\mathrm{p}} = \{\psi_1^p, \psi_2^p \dots \psi_{C_p}^p\}$, which will be fine-tuned with local data to capture task-specific details.

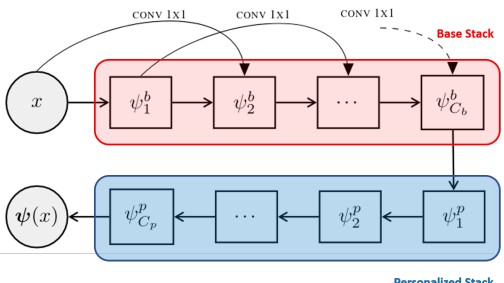

Figure 2: Feature mapping down the component stacks of our architecture space. Every *base cell* takes as inputs (a) the outputs from its immediate predecessor and (b) the one before it through a skip-ahead connection. On the other hand, every *personalization cell* takes as input only the output from the previous cell.

We explain the main difference between these components to account for different level of expressiveness requirements below:

**Base stack.** Every cell $\psi_t^b$ in the base stack takes as inputs the outputs of its previous two cells $\psi_{t-1}^b$ and $\psi_{t-2}^b$ (replaced with raw input $x$ when necessary for $t \le 2$). The output of the skip-ahead cell

$\psi_{t-2}^b$ is additionally passed through a $1 \times 1$ convolution layer as a cost-effective way to control the number of channels. Additionally, the operators available to the base cell include large convolution layers with size $5 \times 5$ and $7 \times 7$. To compensate for the growing number of channels, we periodically employ a reduction convolution (with stride larger than 1) similar to DSNAS (Hu et al., 2020) to reduce the feature dimension down the stack.

**Personalized stack.** As opposed to the design of the base cells above, every cell $\psi_t^p$ in the personalized stack has minimal expressiveness. That is, $\psi_t^p$ excludes large operators and only takes as input the output of its immediate predecessor $\psi_{t-1}^p$ (or $\psi_{C_b}^b$ when $t = 1$). There are two reasons for this choice. First, as the fine-tuning phase has access to fewer data samples than the federated phase, having a more compact fine-tuning space helps to improve the rate of convergence. Second, as we will discuss in Section 3.4 below, our personalized FL objective requires the Hessian of the personalized parameters, which is computationally expensive. As such, we only restrict the personalization to happen on the more compact personalized stack.

### 3.3 PERSONALIZED FEDERATED LEARNING OBJECTIVE

Unlike FEDAVERAGING (McMahan et al., 2017), which assumes the clients will follow the consensus base model obtained after the federated phase, FEDPNAS expects clients to further personalize the base model with local task data. That said, while the base model is trained to work well in the expected sense over the task distribution, there is no guarantee that it is a good initial point for every client model to improve upon via fine-tuning. To address this, we adopt the concept of training in anticipation of future adaptation introduced by MAML (Finn et al., 2017). That is, during client update, instead of optimizing the loss with respect to the same consensus weight, each client will instead optimize the weight perturbed by a small gradient step in the fine-tuning direction.

---

**Algorithm 1** FEDPNAS - FEDERATED PHASE

1: CENTRALAGGREGATION:
2: $\theta_0 \leftarrow$ INITIALIZEPARAMETER
3: **for** $t = 1, 2 \ldots T_s$ **do**
4:     **for** $k = 1, 2 \ldots M$ **in parallel do**
5:         $\theta_t^k \leftarrow$ CLIENTUPDATE$(k, \theta_{t-1})$
6:     $\theta_t \leftarrow \sum_{k=1}^M \frac{1}{M} \theta_t^k$

7: CLIENTUPDATE$(k, \theta)$:
8: **for** $t = 1, 2 \ldots T_c$ **do**
9:     **for** batch $(x, y) \in \mathcal{D}_k$ **do**
10:         $\mathcal{L}_k, \nabla \mathcal{L}_k \leftarrow$ EVAL$(x, y; \theta_b, \theta_p)$
11:         $\tilde{\theta}_p \leftarrow$ GRADUPDATE$(\nabla_{\tilde{\theta}_p} \mathcal{L}_k)$
12:         $\tilde{\mathcal{L}}_k, \nabla \tilde{\mathcal{L}}_k \leftarrow$ EVAL$(x, y; \theta_b, \tilde{\theta}_p)$
13:         $\nabla_{\theta_p} \tilde{\mathcal{L}}_k \leftarrow$ EQ. 5
14:         $\theta_b, \theta_p \leftarrow$ GRADUPDATE$(\nabla_{\theta_b, \theta_p} \tilde{\mathcal{L}}_k)$
15: **return** $\theta$ **to central server**

---

**Algorithm 2** FEDPNAS - EVAL

1: **Input:** $x, y, \theta$
2: SKIP, PREV $\leftarrow x, x$
3: $\mathbf{W}, \mathbf{\Pi} \leftarrow \theta$
4: **for** CELL $\psi \in \boldsymbol{\psi}$ **do**
5:     $\mathbf{Z} \leftarrow$ SAMPLEOPS $(x, \text{PREV}; \mathbf{\Pi})$
6:     $\psi \leftarrow$ EXTRACTCHILDNET $(\mathbf{Z}, \mathbf{W})$

7:     **if** $\psi \in \boldsymbol{\psi}_b$ **then**
8:         OUTPUT $\leftarrow \psi(\text{PREV}, \text{SKIP})$
9:     **else if** $\psi \in \boldsymbol{\psi}_p$ **then**
10:         OUTPUT $\leftarrow \psi(\text{PREV})$
11:     SKIP $\leftarrow$ PREV
12:     PREV $\leftarrow$ OUTPUT
13: $\mathcal{L} \leftarrow$ LOSS(OUTPUT, $y$)
14: $\nabla \mathcal{L} \leftarrow$ BACKPROP$(\mathcal{L})$
15: **return** $\mathcal{L}, \nabla \mathcal{L}$

---

For simplicity, let $\theta = \{\theta_b, \theta_p\}$ respectively denote all trainable parameters of the base stack and the personalized stack, i.e., $\theta_b = \{\mathbf{W}_b, \mathbf{\Pi}_b\}, \theta_p = \{\mathbf{W}_p, \mathbf{\Pi}_p\}$. The personalized FL objective at client $i$ is then given by $\tilde{\mathcal{L}}_i(\theta_b, \theta_p) \triangleq \mathcal{L}_i(\theta_b, \tilde{\theta}_p)$ where $\tilde{\theta}_p \triangleq \tilde{\theta}_p - \eta \nabla_{\theta_p} \mathcal{L}_i(\theta_b, \theta_p)$ adjusts the parameters of the personalized component to account for a small fine-tuning gradient step. The adjusted local loss only depends on the respective client data and is amenable to federated learning. The local update gradient, however, involves a Hessian term whose computation is expensive to repeat over many epochs. To circumvent this problem, we use the first-order Taylor approximation to estimate the Hessian term by the outer product of Jacobian, which results in a gradient that requires exactly two forward/backward passes to compute:

$$
\begin{aligned}
\nabla_{\theta_p} \tilde{\mathcal{L}}_i &= \left( \nabla_{\theta_p} \tilde{\theta}_p \right) \left( \nabla_{\tilde{\theta}_p} \tilde{\mathcal{L}}_i \right) \\
&= \left( \mathbf{I} - \eta \nabla_{\theta_p}^2 \mathcal{L}_i \right) \left( \nabla_{\tilde{\theta}_p} \tilde{\mathcal{L}}_i \right) \\
&\simeq \left( \mathbf{I} - \eta \nabla_{\theta_p}^\top \mathcal{L}_i \nabla_{\theta_p} \mathcal{L}_i \right) \left( \nabla_{\tilde{\theta}_p} \tilde{\mathcal{L}}_i \right)
\end{aligned}
\tag{5}
$$

where $\mathcal{L}_i$ and $\tilde{\mathcal{L}}_i$ are short-hands for $\mathcal{L}_i(\theta_b, \theta_p)$ and $\tilde{\mathcal{L}}_i(\theta_b, \theta_p)$ respectively. The FL phase of our FEDPNAS framework is detailed in Alg. 1. An instance of FEDPNAS's forward and backward pass which sequentially unrolls down the component stacks, alternating between sampling and evaluation, is in turn given in Alg. 2.

## 3.4 CONTEXT-AWARE OPERATOR SAMPLER

The choice of a fully factorizable sampling distribution $p(\mathbf{Z})$ in DSNAS follows that of SNAS (Xie et al., 2018), which argues that the Markov assumption for $p(\mathbf{Z})$ is not necessary because NAS has fully delayed rewards in a deterministic environment. However, this generally only holds for two-stage NAS (Section 2.1) and does not apply to end-to-end frameworks such as SNAS and DSNAS. We instead to take advantage of the over-parameterized architecture via factorizing the conditional $p(\mathbf{Z} \mid x)$, which takes into account the temporal dependency of structural decisions:

$$
\begin{aligned}
p(\mathbf{Z} \mid x) &= p(\mathbf{Z}_1 \mid x) \prod_{t=2}^{C} p(\mathbf{Z}_t \mid \mathbf{Z}_{t-1} \ldots \mathbf{Z}_1, x) \\
&\simeq p(\mathbf{Z}_1 \mid x) \prod_{t=2}^{C} p(\mathbf{Z}_t \mid v_1^t, x) \\
&= p(\mathbf{Z}_1 \mid x) \prod_{t=2}^{C} \prod_{(i,j)} p(\mathbf{Z}_{ij}^t \mid v_1^t, x) ,
\end{aligned}
\tag{6}
$$

where $\mathbf{Z}_t$, $\mathbf{Z}_{ij}^t$ and $v_1^t$ respectively denote all the samples, the sample at edge $(i, j)$ and the input at cell $\psi_t$. We have also assumed a single stack setting since the parameterization of $p(\mathbf{Z})$ does not differ between base and personalized cells.

To reduce computational complexity, instead of conditioning the samples of subsequent cells on previous $\mathbf{Z}$ samples, we approximate $p(\mathbf{Z}_t \mid \mathbf{Z}_{t-1} \ldots \mathbf{Z}_1, x) \simeq p(\mathbf{Z}_t \mid v_1^t, x)$ by the assumption that the cell contents are conditionally independent given the immediate cell input and the original input. Finally, we assume that $p(\mathbf{Z}_t \mid v_1^t, x)$ is fully factorizable across edges in the same cell and parameterize $p(\mathbf{Z}_{ij}^t \mid v_1^t, x) = \phi^{(i,j)}(v_1^t, x)$ where $\phi$ is a deep classification network whose output dimension equal the number of edges in cell $\psi_t$. Samples of $\mathbf{Z}_{ij}^t$ can then be generated using the straight-through Gumbel-softmax reparameterization similar to Jang et al. (2016).

## 3.5 THEORETICAL CONNECTION TO STANDARD GRADIENT UPDATE

Finally, we analyze the connection of our gradient update framework to the standard gradient update, and explain why it is critical in achieving a *vantage* point that improves average objective value without compromising any local objective. First, we note that the gradient update Eq. 5 in Section 3.3 at the $t$-th iteration can be written as:

$$
\begin{aligned}
\theta_p^{t+1} &= \theta_p^t - \frac{\eta_2}{M} \sum_{i=1}^{M} \nabla_{\theta_p} \mathcal{L}_i(\theta_b^t, \tilde{\theta}_{p,i}^t) + \frac{\eta_1 \eta_2}{M} \sum_{i=1}^{M} \alpha_i \nabla_{\theta_p} \mathcal{L}_i(\theta_b^t, \theta_p^t) , \\
\text{and} \quad \theta_b^{t+1} &= \theta_b^t - \frac{\eta_2}{M} \sum_{i=1}^{M} \nabla_{\theta_b} \mathcal{L}_i(\theta_b^t, \tilde{\theta}_{p,i}^t) ,
\end{aligned}
\tag{7}
$$

where $\tilde{\theta}_{p,i}^t$ denotes the $i$-th local personalized parameters; $\eta_1$ and $\eta_2$ are two separate learning rates and $\alpha_i \triangleq \nabla_{\theta_p}^\top \mathcal{L}(\theta_b^t, \theta_{p,i}^t) \nabla_{\theta_{p,i}} \mathcal{L}(\theta_b^t, \tilde{\theta}_{p,i}^t)$ (See Appendix A for detailed derivation). This implies that our federated personalize update corresponds to a federated update scheme with three gradient steps: (1) $\theta_p$ takes a *local* gradient (w.r.t. locally updated parameters) step of size $\eta_1$; (2) Both $\theta_b$ and $\theta_p$ take a *federated* gradient (w.r.t. server-wide parameters averaging) step of size $\eta_2$; (3) $\theta_p$ takes a *weighted federated* gradient step of size $\eta_1 \eta_2$, where the weight of client $i$ is given by $\alpha_i$.

Explicitly, Step 1 and 2 together comprise a special instance of FEDAVERAGING (McMahan et al., 2017), where $\theta_b$ take one gradient step for every two gradient steps taken by $\theta_p$. Step 3, on the other

hand, takes the information of the two gradient steps of $\theta_p$ and adjust the magnitude of the local gradient step (whose direction is given by $\nabla_{\theta_p}\mathcal{L}_i(\theta_b^t, \theta_p^t)$) accordingly to trade-off between preserving local objective value and improving average objective value. We then theorize the scenario in which such an update is beneficial and state the following assumption to lay the foundation of our analysis:

**Assumption 1** *For a fixed instance of $\theta_b$, let $\tilde{\theta}_{p,i} = \theta_{p,i} - \eta\nabla_{\theta_p}\mathcal{L}(\theta_b, \theta_{p,i})$ denote the personalized parameters after a local update step (i.e., step 1 above) at client $i$, then there exists a distribution $\mathcal{S}$ on matrix $\mathbf{S} \in \mathbb{R}^{k \times |\theta_p|}$ that satisfies*

$$\forall \mathbf{x} \in \mathbb{R}^n, \|\mathbf{x}\|_2 = 1 : \mathbb{E}_{\mathbf{S}\sim\mathcal{S}}\left[|\|\mathbf{Sx}\|_2^2 - 1|^\ell\right] \quad \leq \quad \epsilon^\ell \cdot \delta\,, \tag{8}$$

$$\Pr_{\mathbf{S}\in\mathcal{S}}\left(\left|\nabla_{\theta_p}\mathcal{L}(\theta_b, \tilde{\theta}_{p,i}) - \mathbf{S}^\top\mathbf{S}\left(\frac{1}{M}\sum_{i=1}^M \nabla_{\theta_p}\mathcal{L}(\theta_b, \tilde{\theta}_{p,i})\right)\right| \leq \sqrt{\frac{6}{k\delta}}\right) \quad \geq \quad 1 - \delta \tag{9}$$

*where $k = \mathcal{O}\left(C\|\theta_b - \theta_b^*\|_2^{-2}\right)$ for some constant $C > 0$ and $\theta_b^*$ denotes the optimal base parameters.*

The above assumption implies that, as $\theta_b$ improves and better captures the broad similarity across tasks, the local personalized components will diverge to capture the differences. It then becomes less likely for the FEDAVG personalized gradient to capture all these differences simultaneously. That is, suppose there exists an affine transformation to reconstruct the local component $\nabla_{\theta_p}\mathcal{L}_i(\theta_b^t, \tilde{\theta}_p^t)$ from the federated gradient $\mathbf{L} \triangleq \frac{1}{M}\sum_{i=1}^M \nabla_{\theta_p}\mathcal{L}_i(\theta_b^t, \tilde{\theta}_p^t)$, then the rank of this affine transformation would be inversely proportionate to $\|\theta_b - \theta_b^*\|_2$.

Finally, Proposition 1 below shows that when this assumption holds and $\theta_b$ converges to the optimal parameter $\theta_b^*$ (i.e., the error term tends to 0), then with very high probability, the coefficient $\alpha_i$ of the *weighted* federated gradient step (i.e., step 3 above) accurately captures the cosine similarity between the local gradient (step 1) and the federated gradient (step 2).

**Proposition 1** *Suppose assumption 1 holds, then with probability at least $1 - 2\delta$ and normalized gradients, we have:*

$$\left|\alpha_i - \nabla_{\theta_p}^\top\mathcal{L}_i(\theta_b^t, \theta_p^t)\mathbf{L}\right| \quad = \quad \mathcal{O}(\|\theta_b - \theta_b^*\|/\delta) \tag{10}$$

***Proof.*** *See Appendix B*

This result has strong implication with respect to the scenario with multiple heterogeneous tasks, whose local gradients contradict in directions. Per this setting, we expect a standard federated gradient update scheme to encourage parameters drifting in the general direction of the majority (i.e., captured by the federated gradient), thus worsening the performance of tasks that are in the minority. Proposition 1, however, implies that whenever the local gradient contradicts the federated gradient, $\alpha_i$ will be close to the cosine similarity term, which is negative. This in turn results in a dampening effect on the federated gradient and helps to preserve the client performance on its own local task.

## 4 EXPERIMENTS

This section describes our experiments to showcase the performance of FEDPNAS compared to different NAS and FL benchmarks on various scenarios. All of our empirical studies are conducted on two image recognition datasets: (a) the CIFAR-10 dataset (Krizhevsky et al., 2009) which aims to predict image labels from 10 classes given a train/test set of $50000/10000$ colour images of dimension $32 \times 32$ pixels; and (b) the MNIST dataset (LeCun et al., 2010) which aims to predict handwritten digits (i.e. 0 to 9) given a train/test set of $60000/10000$ grayscale images of dimension $28 \times 28$ pixels. Our search space entails $2^{40}$ possible architectures, which is detailed in Appendix D. We compare two variants of our framework, CA-FEDPNAS (with context-aware operation sampler) and FEDPNAS (without the operation sampler), against: (a) FEDAVERAGING of a fixed architecture to justify the need for NAS in FL; (b) FEDDSNAS - the federated extension of DSNAS (Section 3.1) to show the effectiveness of our proposed context-aware sampler on NAS performance; and finally (c) CA-FEDDSNAS, which extends FEDDSNAS with our context-aware sampler.

**On simulate heterogenous predictive tasks.** To simulate this scenario, we first distribute the data i.i.d across clients (10000/2000 and 12000/2000 training/test images per client for CIFAR-10 and MNIST datasets respectively). Then, we independently apply a different transformation to each partitioned dataset. Input images within the same train/test set is subject to the same transformation. In both our experiments, the client datasets are subjected to rotations of $-30°, -15°, 0°, 15°$ and $30°$ respectively. This data generation protocol reflects a realistic and frequently seen scenario where independently collected data of the same phenomenon might contain systematic bias due to measurement errors and/or different collection protocols. Fig. 3 below shows the performance of all the methods in comparison, plotted against number of search epochs and averaged over the above rotated variants of CIFAR-10 and MNIST datasets.

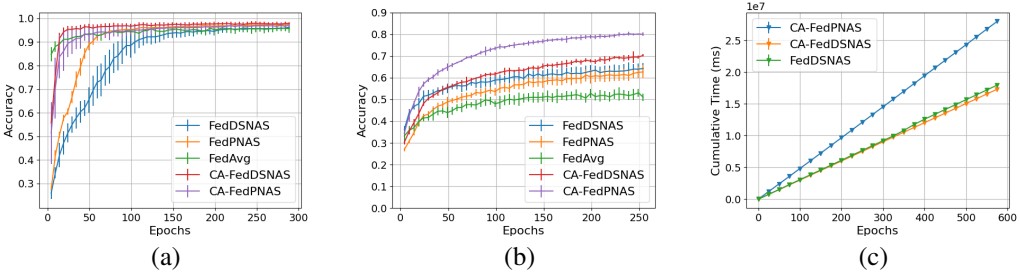

(a)        (b)        (c)

Figure 3: Plotting average classification accuracy of various methods against no. training epochs on heterogeneous tasks derived from (a) MNIST dataset; and (b) CIFAR-10 dataset. Figure (c) compares cumulative running time of various methods against no. training epochs on CIFAR-10 dataset.

On the MNIST dataset (Fig. 3b), all methods eventually converge to a similar performance. Among the NAS benchmarks, FEDPNAS and FEDDSNAS both converge slower than FEDAVG and start off with worse performance in early iterations, which is expected since FEDAVG does not have to search for the architecture and it is likely that the default architecture is sufficient for the MNIST task. On the other hand, we observe that both CA-FEDPNAS and CA-FEDDSNAS converge much faster than their counterparts without the context-aware operation sampler component. This shows that making use of contextual information helps to quickly locate regions of high-performing architectures, especially on similar inputs.

On the CIFAR-10 dataset (Fig. 3a), we instead observe significant gaps between the worst performing FEDAVG and other NAS methods. This is likely because the default architecture does not have sufficient learning capability, which confirms the need for customizing solutions. Among the NAS benchmarks, we again observe that both CA-FEDPNAS and CA-FEDDSNAS outperform their counterparts without our operation sampler, which confirms the intuition above. Most remarkably, our proposed framework CA-FEDPNAS achieves the best performance (0.8) and significantly outperformed both variants of federated DSNAS (0.71 for CA-FEDDSNAS and 0.63 for FEDDSNAS).

Lastly, Fig. 3c shows the runtime comparison between three methods on the CIFAR-10 experiment. In terms of sampling time, we observe that there is negligible overhead incurred by using our context-aware sampler (CA-FEDDSNAS vs. FEDDSNAS). The time incurred by our update (CA-FEDPNAS) scales by a constant factor compared to CA-FEDDSNAS since we use exactly one extra forward/backward pass per update.

**On objectives with varying heterogeneity.** We expand the above study by investigating respective performance of CA-FEDPNAS and FEDDSNAS on tasks with varying levels of heterogeneity. At low level of heterogeneity, we deploy these methods on 5 sets of slightly rotated MNIST images. At high level of heterogeneity, we employ a more diverse set of transformations on MNIST images, such as hue jitter and large angle rotations of $90°$ and $-90°$. Table 1 show the respective result of each task from these two settings. We observe that our method CA-FEDPNAS achieves better performance on most tasks and the performance gaps on tasks with higher heterogeneity are more pronounced (i.e., up to 7% improvement on ROTATE 90 task). This clearly shows the importance of architecture personalization when the training tasks are significantly different and justifies our research goal.

| Heterogeneity | Task Description | FedDSNAS | CA-FedPNAS |
|---|---|---|---|
| Low | Rotate -30 | 0.947 | **0.978** |
| | Rotate -15 | 0.973 | **0.976** |
| | Vanilla | **0.988** | 0.985 |
| | Rotate 15 | 0.986 | **0.987** |
| | Rotate 30 | 0.972 | **0.981** |
| High | HueJitter -0.5 | 0.966 | **0.978** |
| | HueJitter 0.5 | 0.967 | **0.972** |
| | Vanilla | 0.988 | **0.989** |
| | Rotate -90 | 0.892 | **0.932** |
| | Rotate 90 | 0.866 | **0.932** |

Table 1: Predictive accuracy of CA-FedPNAS FedDSNAS on tasks with varying heterogeneity levels. Rotate X denotes a rotation transformation of $X°$ on client data; Vanilla denotes the original MNIST images; and HueJitter X denotes a hue jitter transformation of training images by a factor of X. The best performance in each row is in bold font.

**On knowledge transfer to completely new tasks.** Finally, we investigate a scenario where the architecture distributions discovered by CA-FedPNAS and FedDSNAS are required to generalize to completely unseen tasks. Particularly, we train both methods on five clients whose local data consist of 12000 slightly rotated CIFAR-10 images (i.e., in the range of $\pm 30°$), similar to the setting of the first experiment. During testing, however, we supply each local client with 2000 test images subjected to related but completely unseen transformations (i.e., $90°$ and $-90°$ rotations).

Our results are summarized in Table 2. First, we measure the performance of CA-FedPNAS and FedDSNAS without any weight retraining. When received no additional information from the unseen tasks, both methods perform poorly as expected. While CA-FedPNAS achieves better predictive accuracy, the performance gap in this scenario is negligible. To provide additional clues for adaptation, albeit minimal, we retrain the weights of each local model with 200 images rotated according to respective unseen task description. Here, the parameters of our operator sampler component, (and respectively, FedDSNAS's categorical distribution parameters), are frozen to gauge the quality of the learned architecture distributions. Our results show that, with only 100 retraining iterations on limited data, CA-FedPSNAS already outperforms FedDSNAS (5% and 8% improvement respectively on two unseen tasks). This implies that CA-FedPNAS has more accurately capture the broad similarity of the task spectrum through the personalized architecture distribution, which requires minimal additional information to successfully adapt to unseen tasks.

| Unseen Task Description | FedDSNAS | CA-FedPNAS | FedDSNAS (Retrained) | CA-FedPNAS (Retrained) |
|---|---|---|---|---|
| Rotate -90 | $0.545 \pm 0.04$ | $0.578 \pm 0.09$ | $0.699 \pm 0.12$ | $\mathbf{0.734 \pm 0.17}$ |
| Rotate 90 | $0.553 \pm 0.12$ | $0.569 \pm 0.06$ | $0.673 \pm 0.13$ | $\mathbf{0.727 \pm 0.22}$ |

Table 2: Predictive accuracy (averaged over 5 clients) and standard deviation of CA-FedPNAS and FedDSNAS on two unseen tasks (CIFAR-10). The best performance in each row is in bold font.

## 5 Conclusion

We demonstrate that federated learning for multi-task scenarios requires extensive personalization on the architecture level to obtain good predictive performance. This paper identifies two potential sources of model personalization: (1) task-personalization, which aims to select architectures best suited for specific learning objectives; and (2) context-personalization, which aims to select architectures best suited for specific input samples. To incorporate these aspects of personalization into Federated NAS, we propose FedPNAS which consists of two main components: (1) a context-aware operator sampler which learns a sampling distribution for feature maps along a master architecture; and (2) a personalized federated learning objective which anticipates client fine-tuning and guides the federated model update to regions that tolerate future local updates.

## 6 REPRODUCIBILITY & ETHIC STATEMENT

This work contributes to the literature of Federated Learning through improving the state-of-the-art performance. As such, it could have significant broader impact by allowing users to more accurately solve practical problems. While applications of our work to real data could result in ethical considerations, this is an indirect (and unpredictable) side-effect of our work. Our experimental work uses publicly available datasets to evaluate the performance of our algorithms; no ethical considerations are raised. Our implementation code is published anonymously at https://github.com/icml2021fedpnas/fedpnas. All proofs and details of various architectures are included in the Appendix of this paper.

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
