# OpenReview forum: "Personalized Neural Architecture Search for Federated Learning"
_ICLR.cc/2022/Conference — ICLR 2022 Submitted_

### Official Review · Reviewer_EuZf · 2021-10-31

**Correctness:** 3
**Technical Novelty And Significance:** 2
**Empirical Novelty And Significance:** 2
**Recommendation:** 3
**Confidence:** 4

**Main Review:**

Strength:
- The combination of NAS and FL is mostly novel, there are a few works done but so far all are very recent and unpublished.

Weaknesses:
- Main issue is the experiments. The authors do not compare to any method besides the versions they propose here and fedAvg baseline. Experiments are all very limited as well. The result is that one cannot say anything about the abilities of these models in practice.
- Missing many references to PFL works, and should have compared to at least some of them. A partial list:  Shamsian et al "Personalized federated learning using hypernetworks", Dinh et al "Personalized federated learning with Moreau envelopes", 	Zhang et al "Personalized Federated Learning with First Order Model Optimization", and Collins et al "Exploiting Shared Representations for Personalized Federated Learning".
- We note that in Shamsian et al the authors do investigate having a few models of different sizes to fit varying computational resources. While not the same as NAS it should be referenced as related work.
- The theoretical part is not clear at all. Assumption 1 seems strong and unintuitive, while the meaning of the result in Proposition 1 isn't clear from the text.
- The algorithm isn't clear and so is the discussion about the Hessian that follows. I am pretty sure one can avoid computing the Hessian with efficient Jacobian-vector multiplications that should be easy to compute via backprob, but I am not clear from the text where exactly does the Hessian come from.

**Summary Of The Paper:**

The paper proposes a way to combine neural architecture search (NAS) with federated learning (FL) and personalized federated learning (PFL). The authors describe the basic extension of DSNAS to federated learning and propose some modifications to better suit the federated framework.

**Summary Of The Review:**

The experiments are very lacking, theory and writing is unclear.

---

> ### Author Response · Authors · 2021-11-22
> **Thank you for your recommendation**
>
> 1. Thank you for your suggestion, we will incorporate more benchmarks to demonstrate the effectiveness of our method.
> 2. We will add discussions in the related work section to clearly distinguish our framework with the suggested works. The main contrast is that most existing personalized FL work has not considered personalization on the architecture level, except Shamsian et al. Our work, however, places milder restrictions on the candidate space than Shamsian et al.
> 3. Our analysis relates our personalized gradient step to a combination of standard gradient steps. We showed that under certain conditions, our personalized gradient step helps to prevent a client from being dominated by the ensemble gradient, which helps to retain performance on the local task.
> 4. The Hessian comes from deriving the gradients of the personalized FL objective. We showed the derivation in Eq. (5), where the last step is our approximation to work around it.

---

### Official Review · Reviewer_RES6 · 2021-11-02

**Correctness:** 3
**Technical Novelty And Significance:** 3
**Empirical Novelty And Significance:** 2
**Recommendation:** 5
**Confidence:** 4

**Main Review:**

Positive points:

1.	The paper is well-organized with good motivation on conducting personalized NAS for FL.
2.	The idea of adopting NAS for personalized FL learning seems to be novel and worth explored for researchers and practitioners in both communities.

Concerns:

1.	Lack of experimental results on real dataset to provide the effectiveness of the proposed method. The significance of the experiments is not convincing enough by only using the two very cliché image benchmarks with limited clients.
2.	Some literatures (baselines) are missed such as FedNAS (https://arxiv.org/abs/2004.08546).
3.	The method seems to be a combination of several existing well-known techniques without deliberate narrative and in-depth discussion of why we need this specific design and how they works under diversified settings. For example, I doubt about whether the NAS method could handle the non-iid data situation or stateless setting, which are a common case in FL setting. As even NAS on a single dataset itself is a very difficult problem, more experiments should be conducted along this direction to really show how NAS method applied in the proposed framework performs and affected under different FL settings. Similar discussions should also be considered for the meta-learning algorithms adopted.




**Summary Of The Paper:**

This paper presents a personalized federated learning framework, which leverage a stacked search space of base and personalized architectures along with the context-aware search strategy to learn personalized neural network in the FL scenario. The proposed method is validated on two benchmark image datasets compared to its variants and the FedAvg baseline.

**Summary Of The Review:**

Based on the above concern described above, I currently tend to weakly reject the paper but would like to hear more feedback from the author and discussion from other reviewers towards the final decision.

---

> ### Author Response · Authors · 2021-11-22
> **Thank you for your recommendation**
>
> 1. Thank you for your suggestion, we will incorporate more benchmarks to demonstrate the effectiveness of our method.
> 2. We want to highlight that FedNAS is not about learning personalized architectures. Instead, it focuses on federated learning of the same stochastic architecture. When their training is done, the architecture used across all clients will be similar. On the other hand, we advocate learning a task-specific architecture for each local-client by using context data.
> 3. Task-personalization here can be understood as each client having a slightly different loss function. For example, the loss function of client i is given as $\ell_i(x, y; F) = \ell(F(x), \mathcal{T}_i(y))$ where $F$ is the learned network and $\mathcal{T}_i$ is some transformation. Our paper focuses on dealing with the heterogeneity of local loss functions, rather than the non-iid issue of inputs. For ease of implementation, we chose to implicitly encode this transformation via manipulating the training input given to each client. We understand that this causes confusion, and will clarify this in the final manuscript.

---

### Official Review · Reviewer_MieJ · 2021-11-03

**Correctness:** 3
**Technical Novelty And Significance:** 2
**Empirical Novelty And Significance:** 2
**Recommendation:** 5
**Confidence:** 4

**Main Review:**

Strengths:
+ The authors target the personalization problem in federated learning from a combination of task and context points of view.
+ The authors propose a novel algorithm heavily inspired by DSNAS and adopt it in a federated setting. The algorithm accounts for model heterogeneity by incorporating a base component(trained via federated learning) and a fine tuning component(trained on client data) which is the primary contribution.
+ The paper propose a context aware operator sampler to incorporate context into operator sampling stage of NAS
+ The paper backs up their claims with clear theoretical foundations.

Weakness:
- The paper does not show the variety of architectures that were created due to the added personalization. It would be helpful to see how the personalized architectures differ from each other to judge how well it is really helping accuracy.
- A comparison of the base component of the model learned from federated setting to the model trained on regular NAS with the full data should be present in the paper to get a better idea of how the algorithm performs.
- The search space contains 2^40 possible architecture candidates. The authors should show the search cost as well.



**Summary Of The Paper:**

The paper proposes a personalized neural architecture search technique for federated learning. The paper incorporates both task-personalization and context-personalization. Experimental results on both CIFAR-10 and MNIST datasets demonstrate the promise of the proposed method.

**Summary Of The Review:**

This paper proposes a novel NAS algorithm in a federated setting and backs their claims with theoretical rigor. However, the lack of ablation studies diminishes the merits of the proposed technique.

---

> ### Author Response · Authors · 2021-11-22
> **Thank you for your recommendation**
>
> 1. Due to the fluidity of our architecture selection (i.e., a different set of computation pathways for every batch of input), it is difficult to visualize the personalized architecture. We can, however, show the learned distribution of operators at every edge, and will do so for the final revision.
> 2. We note that the base component serves as an intermediate embedding layer and doesn't directly handle classification. However, to demonstrate this point, we have replaced the personalized component with the static architecture used for FedAvg. This architecture obtained $0.743 \pm 0.025$ and $0.965 \pm 0.039$ accuracy respectively on CIFAR-10 and MNIST datasets. Both are outperformed by CA-FedPNAS as reported in our paper.
> 3. Thank you for your recommendation. We have compared the running time in Fig.1c. The theoretical cost for naively iterating exponentially large search space would be $\mathcal{O}(|A|w)$ where $|A| = 2^{40}$ is the size of the search space, and $w$ is the number of parameters, which is approximately $10^6$ in the context of this paper.

---

> > ### Comment · Reviewer_MieJ · 2021-11-29
> > **Thank you for the response**
> >
> > Thanks the authors for the response. I have read the author's response as well as the comments from other reviewers. Here are my suggestions: It would make the work more informative and convincing to have those visualization includes. The size of the search space is a big concern to me. It would make the paper stronger if the authors could come up with a more efficient way to search through the space to make this work more useful in real-world deployments.

---

### Official Review · Reviewer_1UyE · 2021-11-08

**Correctness:** 3
**Technical Novelty And Significance:** 3
**Empirical Novelty And Significance:** 2
**Recommendation:** 5
**Confidence:** 3

**Main Review:**

Strengths:
1. The algorithm of learning architecture with a base component and a personalized component in FL is novel and interesting.
2. Experiment results are promising, showing that utilizing the context-aware operator sampler significantly improves the performance of both FedPNAS and FedDSNAS.

Weaknesses:
1. Presentation needs to be improved. Also, there are many typos in the manuscript. Fig. 2 is not referred to in the paper.
2. The concept of 'task-personalization' is confusing. What is the difference between 'task-personalization' in this paper and distribution shift across clients? Or client data heterogeneity?
3. The experiment setting is far from a realistic scenario. First, clients' data are not iid distributed. I am curious whether non-iid data distribution will make any difference in the resulting model performance. There are natural federated EMNIST dataset and synthetic federated Cifar10 datasets. Why not use these federated datasets for evaluation? Also, the number of clients participating in the training is 5, which is too small.
4. According to the experiment results in Fig. 3, FedPNAS performs worse than FedDSNAS. Only after adding the context-aware operator sampler (CA), CA-FedPNS outperforms CA-FedDSNAS and FedDSNAS on Cifar-10. Therefore, the experiment results cannot demonstrate the usefulness of the personalized component.

Other comments and questions:
1. It would be interesting to see the accuracy for each client and see if the proposed method is suitable for personalization overall (e.g., all clients' models have high accuracy).
2. According to experiment results, CA is the critical reason for CA-FedPNAS outperforming FedDSNAS. It would be good to see if this conclusion is valid on a more complicated/larger dataset.

**Summary Of The Paper:**

This paper proposes a personalized neural architecture search algorithm (FEDPNAS) for FL. FEDPNAS searches for an architecture with a base component (shared across clients) and a personalized component. It also uses a context-aware operator sampler to learn a sampling distribution for feature maps. It provides a theoretical analysis of the FL objective and empirically demonstrates that FEDPNAS outperforms FedAvg and FedDSNAS over image recognition tasks on CIFAR-10 and MNIST datasets.

**Summary Of The Review:**

The idea of personalized NAS in FL is interesting. The proposed algorithm to learn architecture with a base component and a personalized component is novel. However, the definition of task-personalization is confusing, and the experiment used for evaluation is not sufficient to fully demonstrate the effectiveness of FEDPNAS. Therefore, I think this paper is marginally below the acceptance threshold.

---

> ### Author Response · Authors · 2021-11-22
> **Thank you for your recommendation**
>
> 1. Thanks for your recommendation, we will review the manuscript to get rid of typos.
> 2. Task-personalization here can be understood as each client having a slightly different loss function. For example, the loss function of client i is given as $\ell_i(x, y; F) = \ell(F(x), \mathcal{T}_i(y))$ where $F$ is the learned network and $T_i$ is some transformation. For ease of implementation, we chose to implicitly encode this transformation via manipulating the training input given to each client. We understand that this causes confusion, and will clarify this in the final manuscript.
> 3. As we explained in point 2, the transformation is not meant to generate non iid data, but to capture different tasks. We appreciate your recommendation on using proper datasets to demonstrate this, and will incorporate them in the final manuscript.
> 4. Our personalization method is actually two-fold. To demonstrate the usefulness of the CA component, we only compare with non-CA counterparts. When combined with the personalized learning component, our CA-FedPNAS improves significantly over FedDSNAS.
> 5. We showed individual performance of clients on the transfer learning setting. For other settings, the individual performance of clients behave similarly: Clients with mild data transformation perform better than those with severe data transformations.
> 6. Thanks for your recommendation, we will repeat the experiments on larger datasets.

---

### Decision · Program_Chairs · 2022-01-20

**Decision:**

Reject

**Comment:**

The reviewers had remarkably consistent feedback about this paper. They appreciated the formulation of the federated learning problem with architectures having both shared and private (personalized) components. On the other hand, they felt the experiments were insufficient to prove the effectiveness of the method, and had several suggestions in terms of tasks and datasets. They also felt that it's hard to assess whether the existence of private/personalized components is warranted without visualizing the difference between architectures. Overall, the reviewers had good feedback that could strengthen the paper.